# Ocular and Systemic Manifestations in Paediatric Multisystem Inflammatory Syndrome Associated with COVID-19

**DOI:** 10.3390/jcm10132953

**Published:** 2021-06-30

**Authors:** Tzu-Chen Lo, Yu-Yen Chen

**Affiliations:** 1Department of Medical Education, Taichung Veterans General Hospital, Taichung 407, Taiwan; tzuchenlo@gmail.com; 2School of Medicine, National Yang Ming Chiao Tung University, Taipei 112, Taiwan; 3Department of Ophthalmology, Taichung Veterans General Hospital, Taichung 407, Taiwan; 4Institute of Public Health and Community Medicine Research Center, National Yang Ming Chiao Tung University, Taipei 112, Taiwan; 5School of Medicine, Chung Shan Medical University, Taichung 402, Taiwan

**Keywords:** multisystem inflammatory syndrome, COVID-19, SARS-CoV-2, meta-analysis, conjunctivitis

## Abstract

This study aimed to achieve a better understanding of the epidemiological and clinical characteristics of multisystem inflammatory syndrome in children (MIS-C) following coronavirus disease 2019 (COVID-19). We searched PubMed and Embase between December 2019 and March 2021 and included only peer-reviewed clinical studies or case series. The proportions of patients who had conjunctivitis, systemic symptoms/signs (s/s), Kawasaki disease (KD), and exposure history to suspected/confirmed COVID-19 cases were obtained. Moreover, positive rates of the nasopharyngeal real-time reverse transcriptase polymerase chain reaction (RT-PCR) and serum antibody for severe acute respiratory syndrome coronavirus 2 (SARS-CoV-2) were recorded. Overall, 32 studies with 1458 patients were included in the pooled analysis. Around half of the patients had conjunctivitis. The five most common systemic manifestations were fever (96.4%), gastrointestinal s/s (76.7%), shock (61.5%), rash (57.1%), and neurological s/s (36.8%). Almost one-third presented complete KD and about half had exposure history to COVID-19 cases. The positivity of the serology (82.2%) was higher than that of the nasopharyngeal RT-PCR (37.0%). MIS-C associated with COVID-19 leads to several features similar to KD. Epidemiological and laboratory findings suggest that post-infective immune dysregulation may play a predominant role. Further studies are crucial to elucidate the underlying pathogenesis.

## 1. Introduction

The coronavirus disease 2019 (COVID-19) pandemic, caused by the severe acute respiratory syndrome coronavirus 2 (SARS-CoV-2), induced more than 129 million confirmed cases and 2.8 million deaths globally until the end of March 2021. The affected adults may experience fever, diarrhoea, respiratory failure, and heart/renal failure [1]. Children are less infected and present milder symptoms/signs (s/s) compared to adult patients [2,3,4].

However, the United Kingdom’s National Health Service alerted paediatricians regarding a new outbreak of a Kawasaki-like disease, which was temporally associated with COVID-19 and revealed clinical evidence of SARS-CoV-2 infection [5,6]. Affected children appear to have severe illness, including fever and multi-organ inflammation (shock, cardiac, respiratory, renal, gastrointestinal, or neurological symptoms) similar to the cytokine storm in Kawasaki disease (KD) [7,8,9]. The novel disease is defined as paediatric inflammatory multisystem syndrome temporally associated with SARS-CoV-2 infection (PIMS-TS) or multisystem inflammatory syndrome in children (MIS-C) [10,11,12].

The ocular manifestation (conjunctivitis) is common in MIS-C associated with COVID-19 [7,8]. Retinal manifestations and SARS-Cov-2 detection in retina have also been reported [13,14,15]. Whether conjunctivitis is due to virus attack or related to an inflammatory response remains unknown. Previous studies on adult COVID-19 have detected the presence of SARS-CoV-2 in conjunctiva/tear swab by reverse transcription polymerase chain reaction (RT-PCR), suggesting that the conjunctiva is a transmission route of COVID-19 in adults [1]. Therefore, it is crucial to investigate the relationship between ocular manifestation (conjunctivitis) and COVID-19 in children. Since paediatric cases comprise only 0.8–2.2% of all confirmed cases [3,16], we conducted a meta-analysis to enrol more paediatric patients and increase the statistical power.

The underlying mechanism of MIS-C remains unknown. MIS-C exhibits Kawasaki-like symptoms; hence, it is important to describe its systemic manifestations. Furthermore, exploring the proportion of complete KD among MIS-C patients helps us understand the similarities/differences between MIS-C and KD. Assessing the exposure history to suspected/confirmed cases and analysing the positive rate of virus detection through nasopharyngeal PCR and serology can provide a better explanation of the pathogenesis.

Therefore, we conducted a meta-analysis to investigate the epidemiology and clinical characteristics and the virus testing results among paediatric patients with MIS-C associated with COVID-19, to achieve an in-depth understanding of the novel disease.

## 2. Materials and Methods

### 2.1. Search Strategy

This study was conducted in accordance with the Preferred Items for Systematic Reviews and Meta-Analyses (PRISMA) guidelines [17]. PubMed and Embase were searched for papers published from December 2019 to March 2021, using the keywords “(SARS-CoV-2 OR COVID-19 OR 2019-nCoV) AND (conjunctivitis OR complications OR manifestations) AND (multisystem AND inflammatory OR Kawasaki OR mucocutaneous) AND (child OR paediatric).” Initial screening was performed by examining the titles and abstracts. Candidate papers were further assessed with their full texts to check if they have met the inclusion criteria. Moreover, bibliographies were manually searched for relevant literature.

### 2.2. Inclusion and Exclusion Criteria

Only peer-reviewed journal articles written in English were included in the analysis. Enrolled cases should meet the definition of PIMS-TS or MIS-C provided by the Royal College of Paediatrics and Child Health (RCPCH) [10], World Health Organization (WHO) [11], or Centers for Disease Control and Prevention (CDC) [12]. The studies should be cross-sectional, prospective, or retrospective clinical studies or case series. Studies or case reports with case numbers less than five were excluded. Additionally, reviews, meta-analyses, and conference abstracts were excluded due to possible repeated cases. Two researchers independently assessed the eligibility of these articles. Discrepancies were resolved by a consensus-based discussion.

### 2.3. Extraction of Variables

The following data were recorded from each included article: the first author, date of publication, age (median and range) and gender of participants, total number of patients, the number of patients who had conjunctivitis, systemic manifestations, complete KD, and exposure history to suspected/confirmed COVID-19 cases within 4 weeks. Moreover, we recorded the positive rate of virus detection through nasopharyngeal RT-PCR and serum antibody.

### 2.4. Statistical Analysis

The Comprehensive Meta-Analysis software, version 3 (Biostat, Englewood, NJ, USA) was used for the meta-analysis. The primary outcomes are as follows: (1) the proportion of patients with conjunctivitis and (2) the features and proportions of systemic manifestations. The secondary outcomes included the following: (1) prevalence of complete KD among enrolled cases, (2) proportion of cases having prior exposure to confirmed/suspected COVID-19 cases, (3) the positive rate of RT-PCR from nasopharyngeal samples, and (4) the positive rate of antibody in serology tests. The pooled estimates regarding proportion or rate were calculated by the random-effect method. Forest plots were used to illustrate the prevalence or proportion with a 95% confidence interval (CI).

Furthermore, between-trial heterogeneity was assessed using *I*^2^ statistics. *I*^2^ reveals the extent to which the studies vary due to heterogeneity rather than chance or sampling error. An *I*^2^ statistics of ≥75% represents considerable heterogeneity. Publication bias was determined using Funnel plots and Egger’s test.

## 3. Results

### 3.1. Search Results

Figure 1 illustrates the PRISMA flowchart of study screening and selection. An initial search yielded 464 citations. Among them, 106 were removed because of duplication. After screening the titles/abstracts and full-text examination, 107 non-relevant studies were excluded. In total, 215 studies categorised as reviews, meta-analyses, or case reports with case number less than 5 were also excluded. Furthermore, we excluded another four papers not written in English. Finally, 32 studies with a total of 1458 patients were included in our meta-analysis.

### 3.2. Characteristics of Included Studies

Table 1 summarises the characteristics of the 32 studies [18,19,20,21,22,23,24,25,26,27,28,29,30,31,32,33,34,35,36,37,38,39,40,41,42,43,44,45,46,47,48,49]. Half were conducted in the USA and only three were from Asia. Males constituted 56.4% of the patients in 32 studies. Moreover, in our analysis, the patients in most studies had a median age > 5 years [19,20,21,22,23,24,25,26,27,28,29,30,31,32,33,34,35,36,37,38,39,41,44,46,47,49]. In most studies, the patients being investigated were all COVID-19-related MIS-C cases. However, the studies conducted by Riphagen, Diorio, Verdoni, and Campbell included not only pure MIS-C patients but also COVID-19 cases without MIS-C [27,31,41,45]. We extracted the data only from patients with a definite MIS-C diagnosis. Therefore, the patient number enrolled in our analysis in the four studies was not exactly the same as the total number of patients included in the original studies. 

### 3.3. Primary Outcomes

Table 2 shows the clinical features of the ocular and systemic manifestations of the enrolled patients. The prevalence of conjunctivitis widely varied, from 0% to 93.7%. The common systemic manifestations included fever, gastrointestinal s/s (abdominal pain, diarrhoea, vomiting, etc.), neurological s/s, rash, shock, lymphadenopathy, respiratory s/s, and lip redness.

Figure 2 demonstrates that 48.4% (95% CI, 42.9–54.0%) of the patients had ocular s/s using the random-effects model from pooling of the 32 studies. Figure 3 and Figure 4 display the subgroup analysis according to the five most prevalent systemic s/s: fever, gastrointestinal s/s, shock, rash, and neurological s/s. Using random-effects models, the overall prevalence rates were calculated as 96.4%, 76.7%, 61.5%, 57.1%, and 36.8%, respectively (all *p* < 0.001).

### 3.4. Secondary Outcomes

Table 3 shows the proportion of exposure history, the detection of SARS-CoV-2 by nasopharyngeal RT-PCR and serum antibody, and presentation of complete KD.

Figure 5 shows the overall proportion of complete KD of 30.7% (95% CI, 19.1–45.4%). Figure 6 reveals that 47.4% (95% CI, 34.4–60.8%) of patients had exposure to suspected/confirmed COVID-19 cases. The detection rate of SARS-CoV-2 is presented in Figure 7. The overall positive rate of nasopharyngeal RT-PCR was 37.0% (95% CI, 30.7–43.7%). The serum antibody test achieved a higher positive rate, which was 82.2% (95% CI, 73.8–88.4%).

### 3.5. Heterogeneity and Publication Bias

A moderate heterogeneity in studies was found in evaluating the proportion of patients having conjunctivitis (*I*^2^ = 63.7%). Regarding the proportion of exposure history and complete KD, these studies show high heterogeneities (*I*^2^ = 88.9% and 80.8%, respectively). The funnel plots for calculating publication bias are presented in Figure 8 and were found to be non-significant in the analyses regarding the proportion of conjunctivitis, complete KD, exposure history, positive nasopharyngeal RT-PCR, and positive serum antibody (all of the Egger’s tests revealed *p* > 0.05).

## 4. Discussion

This meta-analysis focused on the ocular and systemic manifestations in paediatric MIS-C patients associated with COVID-19, based on 32 studies derived from the database search. Among the 1458 patients, 48.4% had conjunctivitis. The five most prevalent systemic manifestations were fever (96.4%), gastrointestinal s/s (76.7%), shock (61.5%), rash (57.1%), and neurological s/s (36.8%). Around one-third of the patients had presentations of complete KD and nearly half had exposure history to suspected/confirmed cases. The PCR positive rate of nasopharyngeal PCR was almost 40%, while serum antibody revealed a higher rate of about 80%.

Since the COVID-19 pandemic has spread worldwide, many researchers have noticed the importance of MIS-C, which is a severe complication following SARS-CoV-2 infection in children. Several studies have proposed that SARS-CoV-2 triggers MIS-C [8,50,51,52,53]. The evidence includes the timing of the MIS-C that occurred in several countries related to the SARS-CoV-2 epidemic, the geographical areas matched with SARS-CoV-2 infection, and the positive finding of SARS-CoV-2 or exposure history in MIS-C patients. Furthermore, MIS-C usually occurs one month after SARS-CoV-2, implying that post-infectious inflammation might be the underlying mechanism of MIS-C [54].

MIS-C was first reported as a novel disease that had presentations of multi-organ dysfunction and manifestations resembling KD. KD is among the most prevalent vasculitis in childhood, which classically presented with fever (more than 5 days) and at least four out of five clinical s/s, including conjunctivitis, oropharyngeal mucosa changes, peripheral extremities changes, skin rash, and cervical lymphadenopathy [55]. The mechanisms of KD have long been studied, and several factors have supported the hypothesis that previous infections, mainly those from respiratory viruses, may trigger KD. Although MIS-C and KD have similar clinical manifestations, they still have some differences in several aspects. MIS-C occurs more often in children over 5 years old; however, KD often affects children younger than 5 years old [56,57]. MIS-C is reported more in Western countries, whereas KD is more common in Asian/Pacific regions such as Japan [57]. The frequencies of gastrointestinal symptoms and shock in MIS-C are higher than that in KD [6,58]. In contrast, conjunctivitis and rash are less prevalent in MIS-C compared to KD [6,58]. Furthermore, lab data show that MIS-C usually has more lymphopenia, lower platelet count, higher ferritin level, and higher cardiac biomarkers than KD [53]. Therefore, MIS-C and KD are now regarded as two distinct clinical entities [6,52,53,56,57].

Previous review articles written by Jiang, Kiss, Esposito, and Kabeerdoss et al. have provided a panoramic view of MIS-C [8,51,52,53]. However, they are lacking in precise calculation of the proportion of each s/s, presentation of complete KD, positivity of nasopharyngeal PCR/serum antibody, and exposure history. For a comprehensive understanding of MIS-C, we analysed the data of 1458 patients from 32 studies. To the best of our knowledge, our meta-analysis has included a larger number of studies than previous meta-analyses or reviews. 

Our study revealed that 48.4% of MIS-C patients had conjunctivitis, comparable to a previous meta-analysis conducted by Baradaran, Hoste, Sood, and Aronoff [6,7,59,60]. The prevalence of conjunctivitis in our MIS-C patients was much higher than that in Lofredo’s study of adult COVID-19 patients (around 1.1%) [61]. The fact that the proportion of conjunctivitis is higher in MIS-C than that in adult COVID-19 patients may imply the different pathogenesis of the two diseases. Conjunctivitis in MIS-C might be induced more by systemic immunological reaction than by a local virus attack [53]. Because immunological reaction can last longer, conjunctivitis may have more chances to be detected. 

The five most prevalent systemic manifestations in our MIS-C patients were fever (96.4%), gastrointestinal s/s (76.7%), shock (61.5%), rash (57.1%), and neurological s/s (36.8%). Several meta-analyses supported our findings, with similar prevalence [6,7,8,52,59,60,62]. It is worth noting that the frequencies of gastrointestinal s/s and rash in MIS-C were higher than those in typical adult COVID-19 [52,63,64], demonstrating possible differences in the underlying mechanism between MIS-C and adult COVID-19. 

In our meta-analysis, 30.7% of MIS-C patients had complete KD. A previous study by Jiang et al. found the rate of complete KD to be around 41.1% [8]. It is worth noting that we have only calculated the ratio of complete KD. If we added incomplete KD into the analysis, the proportion should be higher than 30.7%. Furthermore, we found that MIS-C was slightly more prevalent in males than in females, which was consistent with previous meta-analyses or population-based studies [6,7,50,52,59,65]. Kabeerdoss et al., in their review article, demonstrated that there is no significant male predominance in MIS-C [53]. However, previous studies found a stronger male-predominant pattern in KD [66,67]. At present, we do not fully understand why the proportion of males in KD is higher than that in MIS-C. Further genetic, immunological, or epidemiological studies are warranted to investigate this issue.

Our study, as well as several previous studies, found that MIS-C patients had a less positive rate of SARS-CoV-2 in PCR than in serology (mostly IgG), suggesting that this inflammatory syndrome is not mediated by direct viral invasion but coincides with the development of acquired immune responses to SARS-CoV-2 [6,7,8,50,59,68].

Our study revealed that 47.4% of MIS-C patients had a history of contact with COVID-19 cases. The included studies showed that the exposure history ranges from 10.3% to 99.3%, which might be affected by lifestyle. Due to the full-blown and worldwide spread of SARS-CoV-2, the CDC recommends wearing masks and maintaining social distancing when in public. Moreover, hand hygiene and staying at home were introduced as effective ways of infection prevention.

Current treatment of MIS-C mimics the treatment of KD. Guidelines published by the American College of Rheumatology suggested an IVIG dosage of 2 gm/kg and adding methylprednisolone IV 1–2 mg/kg/day for hemodynamic unstable MIS-C or refractory MIS-C. Aspirin should be considered based on clinical condition and risk–benefit evaluation [52,69]. The limitation of this meta-analysis is that we have only focused on the part of epidemiological and clinical characteristics, but did not analyse the part of lab data, treatment, and prognosis.

All the main outcomes of our study did not show significant public bias. However, outcomes regarding conjunctivitis, exposure history, and complete KD demonstrated moderate to high heterogeneities. The high heterogeneities can be explained by the fact that the current inclusion criteria of MIS-C are too broad and might overlap children with different diseases such as KD, toxic shock syndrome (TSS), and macrophage-activation syndrome (MAS). Or it may result from differences in ethnicity, pre-existing health conditions, socioeconomic factors, and access to healthcare. In addition, different timings of laboratory tests and discrepancies in thresholds for defining abnormal values increase the heterogeneity between studies. Another limitation of our analyses is that data for a few variables were not reported in every included study, making direct comparisons between studies difficult. In addition, we did not investigate the genetic variants of SARS-CoV-2 in our analyses. We have tried to add the keywords United Kingdom variant (20I/501Y) and South African variant (20H/501Y.V2) into our previous searching term. However, there have been no studies regarding these genetic variants in children associated with COVID-19. More studies with complete data are warranted to derive a profound knowledge of MIS-C in COVID-19.

## 5. Conclusions

Our study used meta-analysis to investigate the epidemiological and clinical characteristics of the novel MIS-C associated with COVID-19. We found that the common presentations of MIS-C patients include conjunctivitis (48.4%), fever (96.4%), gastrointestinal s/s (76.7%), shock (61.5%), rash (57.1%), and neurological s/s (36.8%), which also occur frequently in KD. However, we also found some clues indicating that MIS-C and KD are distinct in pathogenesis. The finding that 50% of our MIS-C patients had a history of contact with COVID-19 cases demonstrated the importance of hand hygiene and social distancing. Our study reminds physicians of the possibility of MIS-C if children have the above-mentioned manifestations. Further clinical and basic studies will shed light on the predisposing factors, early recognition, and multispecialty care/treatment of this novel paediatric disease associated with COVID-19.

## Figures and Tables

**Figure 1 jcm-10-02953-f001:**
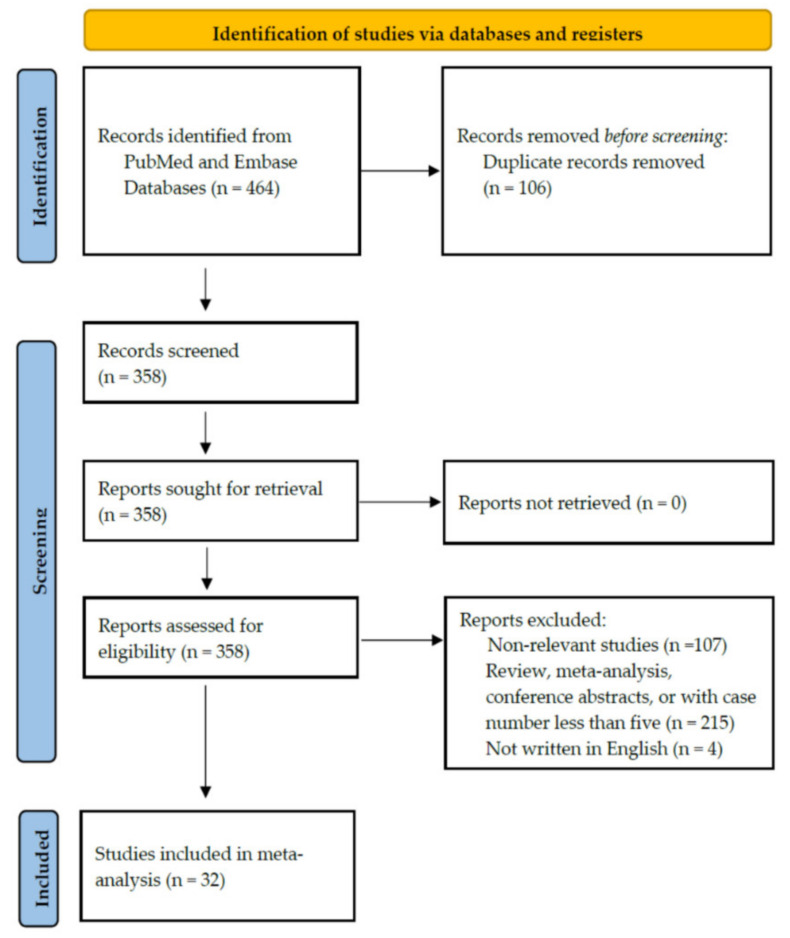
Study retrieval process according to the Preferred Reporting Items for Systematic Reviews and Meta-Analyses (PRISMA) statements.

**Figure 2 jcm-10-02953-f002:**
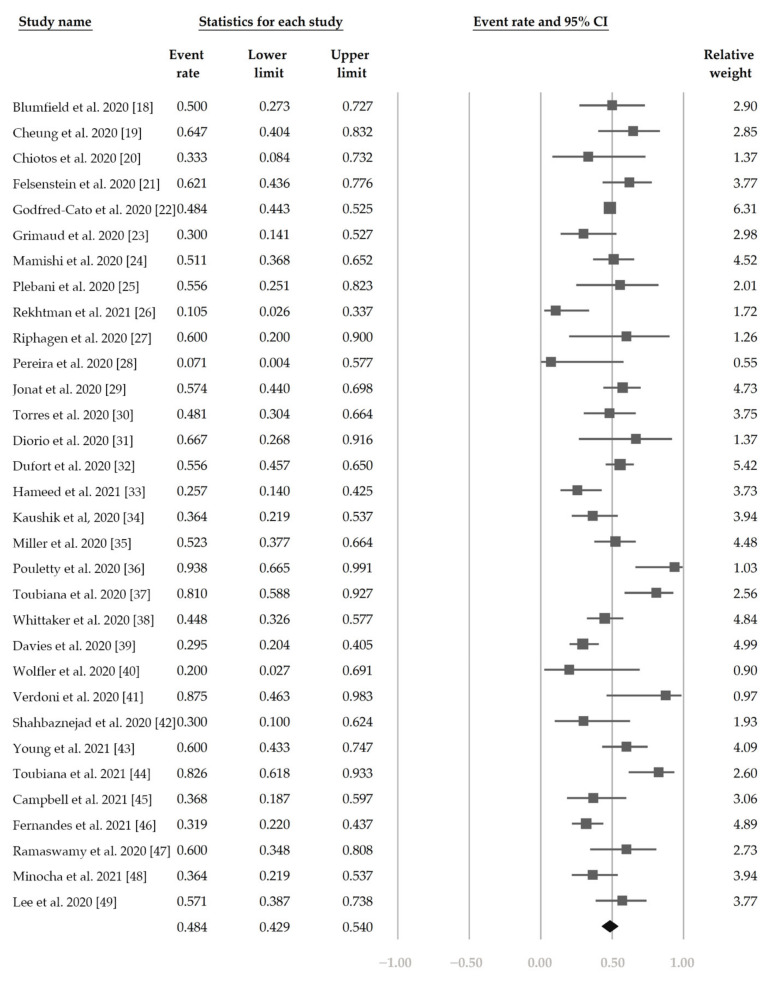
The overall prevalence of conjunctivitis in the included studies.

**Figure 3 jcm-10-02953-f003:**
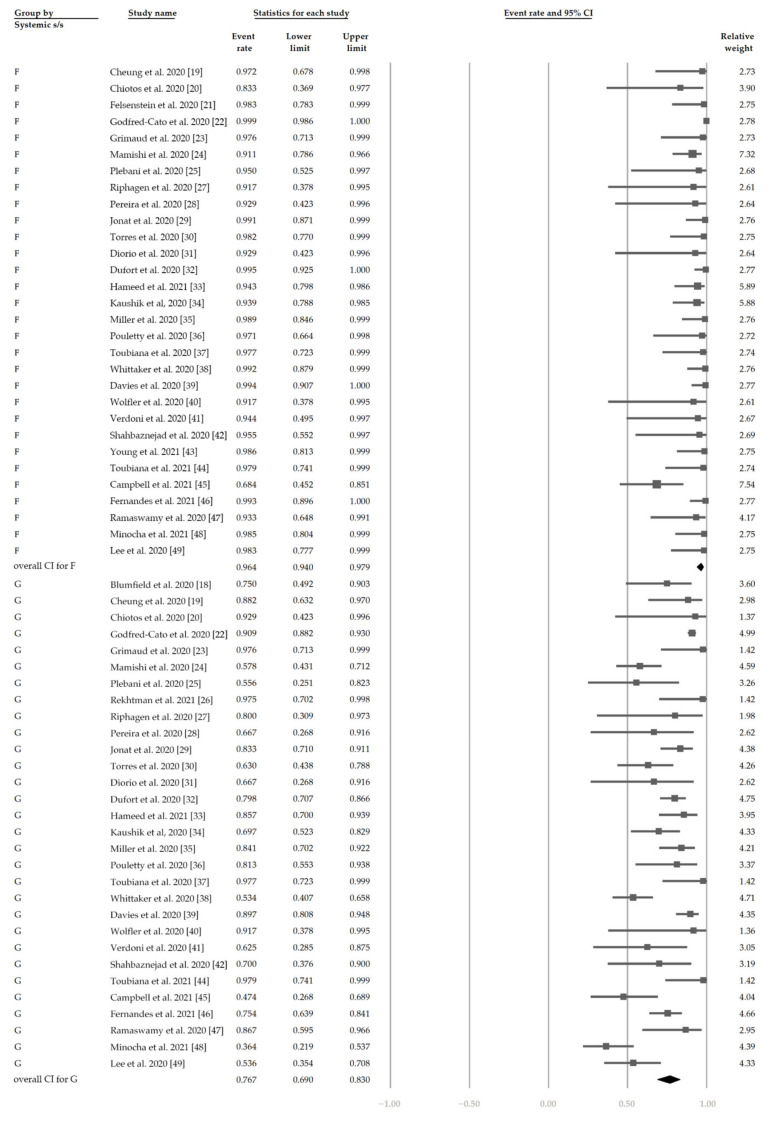
The overall prevalence of fever and gastrointestinal symptoms in the included studies.

**Figure 4 jcm-10-02953-f004:**
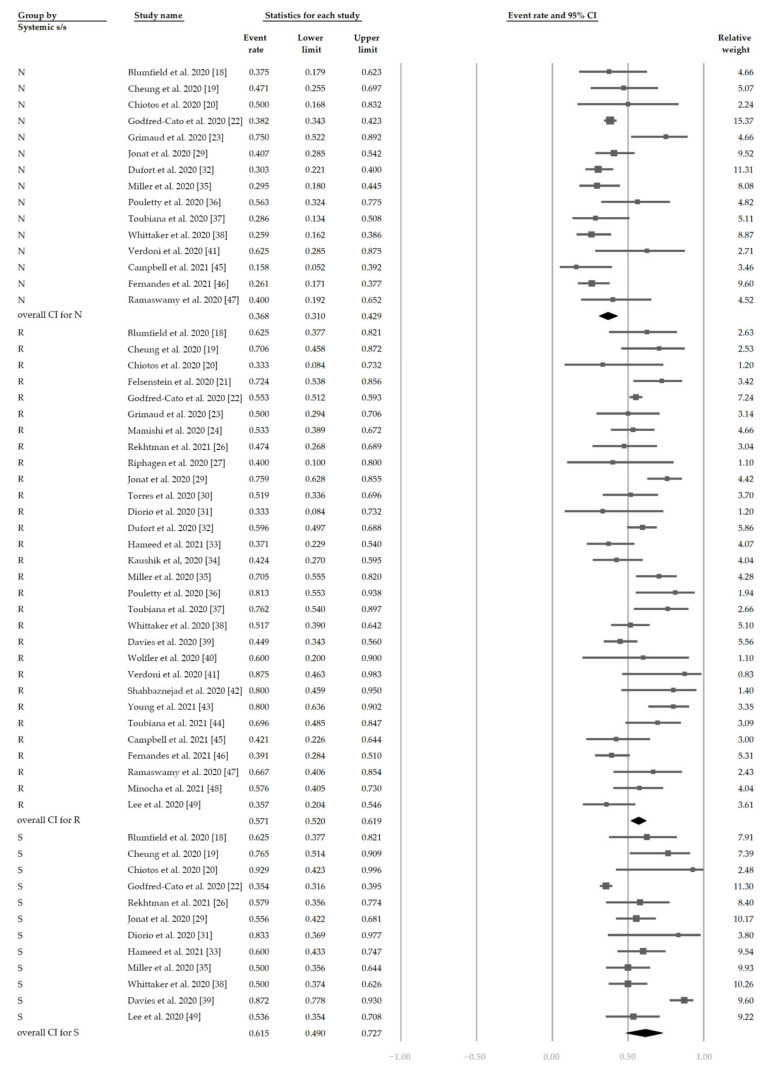
The overall prevalence of neurological symptoms, rash, and shock in the included studies.

**Figure 5 jcm-10-02953-f005:**
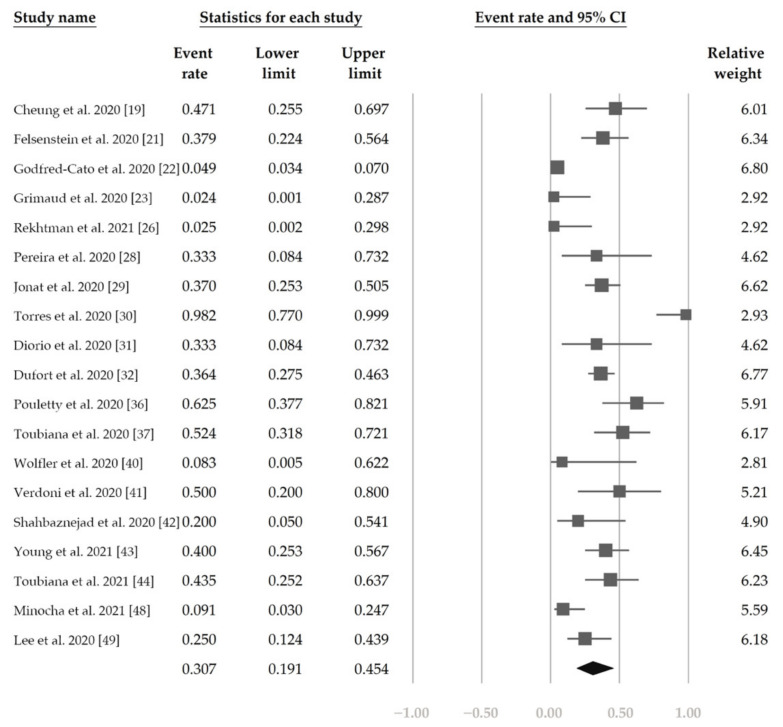
The overall proportion of complete Kawasaki disease.

**Figure 6 jcm-10-02953-f006:**
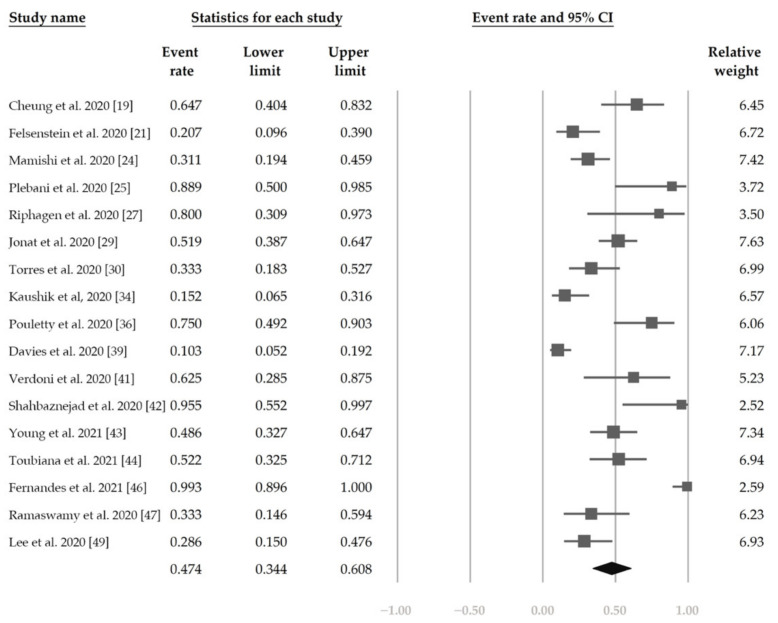
The overall proportion of exposure to suspected/confirmed COVID-19 cases.

**Figure 7 jcm-10-02953-f007:**
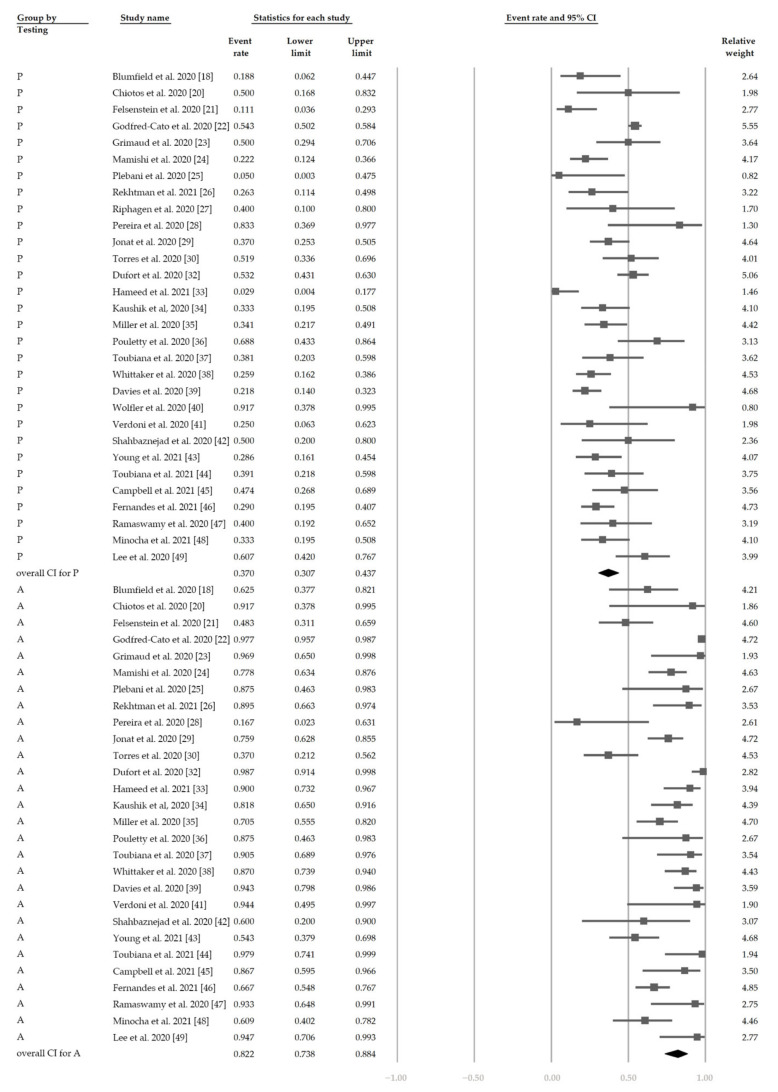
The overall positive rate of SARS-CoV-2. “P” denotes nasopharyngeal RT-PCR (+). “A” denotes serum antibody (+).

**Figure 8 jcm-10-02953-f008:**
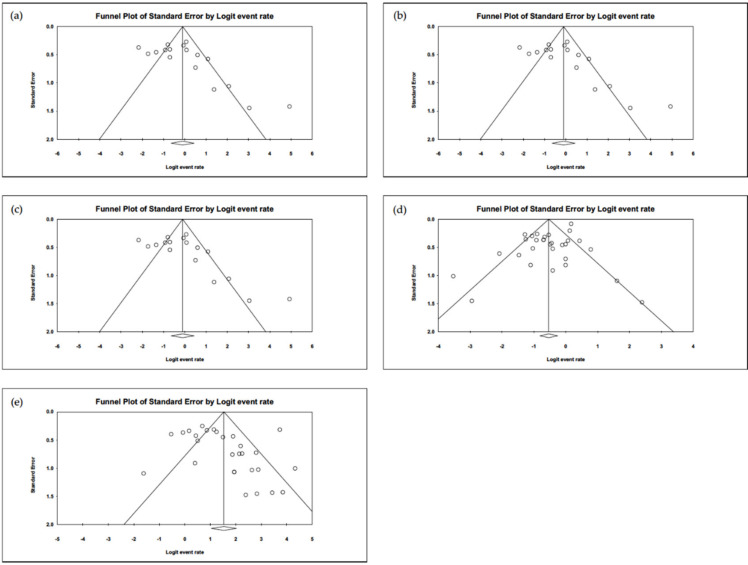
The funnel plots of (**a**) conjunctivitis, (**b**) complete Kawasaki disease, (**c**) exposure history, (**d**) RT-PCR, and (**e**) serum antibody. All the Egger’s tests are shown to be non-significant (*p* > 0.05).

**Table 1 jcm-10-02953-t001:** Demographic characteristics of patients in the studies included in the meta-analysis.

First Author	Type	Publication Date	Country	Pts, *n*	No. of Eyes	Median Age (IQR), Year	Range of Age, Year	Male, *n* (%)	Hospitalized Days †
Blumfield et al. 2020 [18]	R	Aug, 2020	USA	16	32	9.2 ± 4.9 *	1.7–20	10 (63)	NR
Cheung et al. 2020 [19]	P	Jun, 2020	USA	17	34	8(4.9–12)	1.8–16	8 (47)	13–18
Chiotos et al. 2020 [20]	CS	May, 2020	USA	6	12	7.5 (5–12)	5–14	1 (17)	8–17
Felsenstein et al. 2020 [21]	R	Oct, 2020	UK	29	58	6 (3.8–9.9)	NR	20 (69)	14
Godfred-Cato et al. 2020 [22]	C	Aug, 2020	USA	570	1140	8 (4–12)	NR	316 (55)	6
Grimaud et al. 2020 [23]	R	Jun, 2020	France	20	40	8.6 (7–11.6)	2.9–15	10 (50)	1–10
Mamishi et al. 2020 [24]	R	Aug, 2020	Iran	45	90	7 (4–9.9)	0.8–17	24 (53)	8 (6–11)
Plebani et al. 2020 [25]	CS	Oct, 2020	Italy	9	18	10 (3–11)	1.1–14	6 (67)	3–15
Rekhtman et al. 2021 [26]	P	Dec, 2020	USA	19	38	9 (8.5–11.5)	NR	13 (68)	7 (5–10)
Riphagen et al. 2020 [27]	CS	May, 2020	UK	5	10	8 (6–11)	6–14	3 (60)	3–7
Pereira et al. 2020 [28]	CS	Aug, 2020	Brazil	6	12	7.8 (3.9–12.7)	0.01–17.6	5 (83)	NR
Jonat et al. 2020 [29]	P	Oct, 2020	USA	54	108	7 (3.9–13.5)	0.8–20	25 (46)	1–19
Torres et al. 2020 [30]	CS	Aug, 2020	Chile	27	54	6 (3–10)	0–14	14 (52)	9 (6–13)
Diorio et al. 2020 [31]	P	Jul, 2020	USA	6	12	6 (5–7)	5–13	2 (33)	NR
Dufort et al. 2020 [32]	R	Jul, 2020	USA	99	198	9 (4.0–13.3)	NR	53 (54)	6 (4–9)
Hameed et al. 2021 [33]	R	Jun, 2020	UK	35	70	11 (6–14)	NR	27 (77)	25
Kaushik et al, 2020 [34]	R	Jun, 2020	USA	33	66	10 (6–13)	0.2–20	20 (61)	7.8 (6.0–10.1)
Miller et al. 2020 [35]	R	Jun, 2020	USA	44	88	7.3 (4.0–13.7)	0.6–20	20 (45)	26
Pouletty et al. 2020 [36]	R	Jun, 2020	France	16	32	10 (4.7–12.5)	NR	8 (50)	21 (21–24)
Toubiana et al. 2020 [37]	R	Jun, 2020	France	21	42	7.9 (5.8–12.3)	3.7–16.6	9 (43)	5–17
Whittaker et al. 2020 [38]	P	Jun, 2020	England	58	116	9 (5.7–14)	NR	38 (66)	NR
Davies et al. 2020 [39]	R	Jul, 2020	England	78	156	11 (8–14)	NR	52 (67)	5.0 (3.0–6.5)
Wolfler et al. 2020 [40]	R	Jun, 2020	Italy	5	10	NR	0.1–14	NR	5–10
Verdoni et al. 2020 [41]	P	May, 2020	Italy	8	16	6.3 (5.3–7.6)	2.9–9.2	5 (63)	NR
Shahbaznejad et al. 2020 [42]	P	Nov, 2020	Iran	10	20	5.4 ± 3.9 *	1.1–12	6 (60)	NR
Young et al. 2021 [43]	R	Dec, 2020	USA	35	70	2 (1.1–9.5)	0.2–17	19 (54)	2–55
Toubiana et al. 2021 [44]	R	Jan, 2021	France	23	46	8.2 (7.4–10)	NR	12 (52)	NR
Campbell et al. 2021 [45]	R	Nov, 2020	USA	19	38	NR	NR	9 (47)	5–57
Fernandes et al. 2021 [46]	R	Nov, 2020	USA	69	138	7 (3–11)	NR	42 (60)	6 (3–8)
Ramaswamy et al. 2020 [47]	P	Dec, 2020	USA	15	30	10.4 (2.6–18)	NR	8 (53)	NR
Minocha et al. 2021 [48]	R	Sep, 2020	USA	33	66	2.8 (1.4–9)	NR	19 (58)	120
Lee et al. 2020 [49]	R	Jul, 2020	USA	28	56	9 (4.5–13)	0.1–17	16 (57)	1–10

*n*, number; Pts, patients; IQR, interquartile range; R, retrospective; P, prospective; CS, cross-sectional; NR, not reported. † Hospitalized days presented in range or median (IQR). * Mean ± SD.

**Table 2 jcm-10-02953-t002:** Manifestations and clinical characteristics of patients in the included studies.

	Pts, *n*	Conjunctivitis, *n* (%)	Fever, *n* (%)	GI s/s, *n* (%)	Rash, *n* (%)	Shock, *n* (%)	Neurological s/s, *n* (%)	Respiratory s/s, *n* (%)	Lips and Oral Changes, *n* (%)	Cervical Lymphadenopathy, n (%)	Cardiovascular, *n* (%)
Blumfield et al. 2020 [18]	16	8 (50.0)	NR	12 (75)	10 (63)	10 (63)	6 (38)	NR	5 (31)	NR	NR
Cheung et al. 2020 [19]	17	11 (64.7)	17 (100)	15 (88)	12 (70)	13 (76)	8 (47)	7 (41)	9 (53)	6 (35)	NR
Chiotos et al. 2020 [20]	6	2 (33.3)	5 (83)	6 (100)	2 (33)	6 (100)	3 (50)	4 (67)	NR	NR	NR
Felsenstein et al. 2020 [21]	29	18 (62.1)	29 (100)	NR	21 (72)	NR	NR	6 (21)	NR	NR	25 (86)
Godfred-Cato et al. 2020 [22]	570	276 (48.4)	570 (100)	518 (90)	315 (55)	202 (36)	218 (38)	359 (63)	NR	NR	493 (86)
Grimaud et al. 2020 [23]	20	6 (30.0)	20 (100)	20 (100)	10 (50)	NR	15 (75)	NR	NR	NR	NR
Mamishi et al. 2020 [24]	45	23 (51.1)	41 (91)	26 (58)	24 (53)	NR	NR	NR	NR	9 (20)	NR
Plebani et al. 2020 [25]	9	5 (55.6)	9 (100)	5 (56)	NR	NR	NR	3 (33)	NR	NR	NR
Rekhtman et al. 2021 [26]	19	2 (10.5)	NR	19 (100)	9 (47)	11 (58)	NR	11 (58)	NR	NR	NR
Riphagen et al. 2020 [27]	5	3 (60.0)	5 (100)	4 (80)	2 (40)	NR	NR	NR	NR	NR	NR
Pereira et al. 2020 [28]	6	0 (0)	6 (100)	4 (67)	NR	NR	NR	5 (83)	NR	NR	NR
Jonat et al. 2020 [29]	54	31 (57.4)	54 (100)	45 (83)	41 (76)	30 (56)	22 (41)	12 (22)	20 (37)	16 (30)	NR
Torres et al. 2020 [30]	27	13 (48.1)	27 (100)	17 (63)	14 (52)	NR	NR	7 (26)	11 (41)	NR	NR
Diorio et al. 2020 [31]	6	4 (66.7)	6 (100)	4 (67)	2 (33)	5 (83)	NR	4 (67)	NR	NR	NR
Dufort et al. 2020 [32]	99	55 (55.6)	99 (100)	79 (80)	59 (60)	NR	30 (30)	40 (40)	27 (27)	6 (6)	NR
Hameed et al. 2021 [33]	35	9 (25.7)	33 (94)	30 (86)	13 (37)	21 (60)	NR	NR	NR	NR	NR
Kaushik et al, 2020 [34]	33	12 (36.4)	31 (94)	23 (67)	14 (42)	NR	NR	11 (33)	NR	NR	NR
Miller et al. 2020 [35]	44	23 (52.3)	44 (100)	37 (84)	31 (70)	22 (50)	13 (30)	11 (25)	23 (52)	NR	22 (67)
Pouletty et al. 2020 [36]	16	15 (93.7)	16 (100)	13 (81)	13 (81)	NR	9 (56)	2 (13)	14 (88)	6 (38)	NR
Toubiana et al. 2020 [37]	21	17 (81.0)	21 (100)	21 (100)	16 (76)	NR	6 (29)	NR	16 (76)	12 (57)	NR
Whittaker et al. 2020 [38]	58	26 (44.8)	58 (100)	31 (53)	30 (52)	29 (50)	15 (26)	12 (21)	17 (29)	9 (24)	NR
Davies et al. 2020 [39]	78	23 (29.5)	78 (100)	70 (90)	35 (45)	68 (87)	NR	NR	NR	NR	NR
Wolfler et al. 2020 [40]	5	1 (20.0)	5 (100)	5 (100)	3 (60)	NR	NR	3 (60)	NR	NR	NR
Verdoni et al. 2020 [41]	8	7 (87.5)	8 (100)	5 (63)	7 (88)	NR	5 (63)	NR	NR	NR	NR
Shahbaznejad et al. 2020 [42]	10	3 (30.0)	10 (100)	7 (70)	8 (80)	NR	NR	8 (80)	4 (40)	NR	NR
Young et al. 2021 [43]	35	21 (60.0)	35 (100)	NR	28 (80)	NR	NR	NR	17 (49)	NR	NR
Toubiana et al. 2021 [44]	23	19 (82.6)	23 (100)	23 (100)	16 (70)	NR	NR	NR	17 (74)	14 (61)	NR
Campbell et al. 2021 [45]	19	7 (36.8)	13 (68)	9 (20)	8 (42)	NR	3 (16)	14 (74)	NR	NR	18 (95)
Fernandes et al. 2021 [46]	69	22 (31.9)	69 (100)	52 (75)	27 (39)	NR	18 (26)	24 (35)	NR	NR	NR
Ramaswamy et al. 2020 [47]	15	9 (60.0)	14 (93)	13 (87)	10 (67)	NR	6 (40)	NR	NR	NR	NR
Minocha et al. 2021 [48]	33	12 (36.4)	33 (100)	12 (36)	19 (58)	NR	NR	NR	NR	8 (24)	NR
Lee et al. 2020 [49]	28	16 (57.1)	28 (100)	15 (54)	10 (36)	15 (54)	NR	NR	7 (25)	NR	NR

*n*, number; Pts, patients; s/s, symptoms/signs; GI, gastrointestinal; NR, not reported.

**Table 3 jcm-10-02953-t003:** Exposure history, detection of SARS-CoV-2, and presentation of Kawasaki disease.

First Author	Number of Patients	With Complete Kawasaki Disease	Exposure History, * *n* (%)	Nasopharyngeal RT-PCR (+)	Serum Antibody (+)
Blumfield et al. 2020 [18]	16	NR	NR	3/16	10/16
Cheung et al. 2020 [19]	17	8 (47.1%)	11 (64.7%)	8/unknown	9/unknown
Chiotos et al. 2020 [20]	6	NR	NR	3/6	5/5
Felsenstein et al. 2020 [21]	29	11 (37.9%)	6 (20.7%)	3/27	14/29
Godfred-Cato et al. 2020 [22]	570	28 (4.9%)	NR	302/556	418/428
Grimaud et al. 2020 [23]	20	0	NR	10/20	15/15
Mamishi et al. 2020 [24]	45	NR	14 (31.1%)	10/45	35/45
Plebani et al. 2020 [25]	9	NR	8 (88.9%)	0 (0%)	7/8
Rekhtman et al. 2021 [26]	19	0 (0%)	NR	5/19	17/19
Riphagen et al. 2020 [27]	5	NR	4 (80.0%)	2/5	NR
Pereira et al. 2020 [28]	6	2 (33.3%)	NR	5/6	1/6
Jonat et al. 2020 [29]	54	20 (37.0%)	28 (51.9%)	20/54	41/54
Torres et al. 2020 [30]	27	27 (100%)	9 (33.3%)	14/27	10/27
Diorio et al. 2020 [31]	6	2 (33.3%)	NR	5/unknown	1/unknown
Dufort et al. 2020 [32]	99	36 (36.4%)	NR	50/94	76/77
Hameed et al. 2021 [33]	35	NR	NR	1/35	27/30
Kaushik et al. 2020 [34]	33	NR	5 (15.2%)	11/33	27/33
Miller et al. 2020 [35]	44	NR	NR	15/44	31/44
Pouletty et al. 2020 [36]	16	10 (62.5%)	12 (75.0%)	11/16	7/8
Toubiana et al. 2020 [37]	21	11 (52.4%)	NR	8/21	19/21
Whittaker et al. 2020 [38]	58	NR	NR	15/58	40/46
Davies et al. 2020 [39]	78	NR	8 (10.3%)	17/78	33/35
Wolfler et al. 2020 [40]	5	0 (0%)	NR	5/5	NR
Verdoni et al. 2020 [41]	8	4 (50.0%)	5 (62.5%*)*	2/8	8/8
Shahbaznejad et al. 2020 [42]	10	2 (20.0%)	10 (100%)	4/8	3/5
Young et al. 2021 [43]	35	14 (40.0%)	17 (48.6%)	10/35	19/35
Toubiana et al. 2021 [44]	23	10 (43.5%)	12 (52.2%)	9/23	23/23
Campbell et al. 2021 [45]	19	NR	NR	9/19	13/15
Fernandes et al. 2021 [46]	69	NR	69 (100%)	20/69	46/69
Ramaswamy et al. 2020 [47]	15	NR	5 (33.3%)	6/15	14/15
Minocha et al. 2021 [48]	33	3 (9.1%)	NR	11/33	14/23
Lee et al. 2020 [49]	28	7 (25.0%)	8 (28.6%)	17/28	18/19

* Exposure to suspected/confirmed COVID-19 cases within 4 weeks. NR, not reported.

## Data Availability

The data analysed in this study were a re-analysis of existing data, which are openly available at locations cited in the reference section.

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
