# Peer review of "Ocular and Systemic Manifestations in Paediatric Multisystem Inflammatory Syndrome Associated with COVID-19"

_jcm, 2021, doi:10.3390/jcm10132953_

Round 1

Reviewer 1 Report

The authors aimed to achieve a better understanding on the epidemiological and clinical characteristics of multisystem inflammatory syndrome in children (MIS-C) following coronavirus disease (COVID-19). We searched PubMed and Embase between December 2019 to March 2021 and included only peer-reviewed clinical studies or case series. And they found that Epidemiological and laboratory findings suggest that post-infective immune dysregulation may play a predominant role. Further studies are crucial to elucidate the underlying pathogenesis.

This systematic review and meta-analysis are a really well written paper that resume the ocular and systemic manifestation. Some briefs issue must be address prior to continue with publication process.

  1. in the flowchart diagram the excluded studies were 251? Revised this part of the chart?
  2. the PRISMA flowchart have other visual appearance, please re-done
  3. Reference the studies in line 128, correct though the whole manuscript
  4. Table 1, month and day of publication it is not necessary
  5. Table 1, include follow-up of the patients.
  6. Table 1, included number of eyes within the number of patients for eye studies
  7. Table 1, when IQR was not report the SD should be reported, revise this data
  8. Table 2, if systemic sing were repeated within different studies, please use columns to include the sign and then the percent in the row and a different column for other, but visual appearance of the table must be improved.
  9. Figure 2, 3, 5, 6 and 7 could the letter and size change to harmonized with the paper requirements?
  10. and funnel plot could be presented in a 4 part mixed figure to improve to space used
  11. Rewrite third and fourth paragraph of discussion
  12. Conclusion must answer the title and review purpose, change in order to clarify

Reviewer 2 Report

Thank you for your manuscript. Your skill and hard work is evident. The manuscript is clearly written and easy to follow. The authors tackle an important problem related to epidemiological and clinical manifestations of MIS-C following COVID-19. This meta-analysis study will clarify the data about MIS-C and Kawasaki disease.

There are some matters that must be addressed.

Line 49. The authors must include some studies about retinal manifestations of COVID-19. For example: Detection of SARS-CoV-2 in Human Retinal Biopsies of Deceased COVID-19 Patients, Ocular Immunology and Inflammation, 28:5, 721-725, DOI: 10.1080/09273948.2020.1770301

On the other hand the author must explain why they have not used in the “Search strategy” genetic variants (include the so-called United Kingdom variant (20I/501Y), the Brazilian variant (20J/501Y.V3), and the South African variant (20H/501Y.V2).

Reviewer 3 Report

Comments to Authors:

The topic of the review is important and pertaining to the current pandemic crisis worldwide. The authors summarize and discuss various aspects of multisystem inflammatory syndrome following human COVID-19 infection in children including epidemiological and clinical mechanisms underlying MIS-C. Given the quantity of previous studies and reviews on this topic since the beginning of last year (including comparative immune response studies to SARS-CoV-2 in children with that in adults), the authors might want to state more clearly what previous reviews on MIS-C after COVID-19 infection currently exist in the literature, and highlight how their review is intended to uniquely contribute to the understanding AND/OR management of COVID-19 associated MIS-C  in comparison to a few recently published review articles listed in pubmed as follows:

Kiss A, Ryan PM, Mondal T. Management of COVID-19-associated multisystem inflammatory syndrome in children: A comprehensive literature review [published online ahead of print, 2021 Apr 9]. Prog Pediatr Cardiol. 2021;101381. doi: 10.1016/j.ppedcard.2021.101381

Esposito S, Principi N. Multisystem Inflammatory Syndrome in Children Related to SARS-CoV-2. Paediatr Drugs. 2021;23(2):119-129. doi:10.1007/s40272-020-00435-x

Kabeerdoss J, Pilania RK, Karkhele R, Kumar TS, Danda D, Singh S. Severe COVID-19, multisystem inflammatory syndrome in children, and Kawasaki disease: immunological mechanisms, clinical manifestations and management. Rheumatol Int. 2021;41(1):19-32. doi:10.1007/s00296-020-04749-4

Jiang L, Tang K, Levin M, et al. COVID-19 and multisystem inflammatory syndrome in children and adolescents. Lancet Infect Dis. 2020;20(11): e276-e288. doi:10.1016/S1473-3099(20)30651-4

In fact, it does seem apparent that the relevant literature and data is large, diverse and unwieldy without much interpretation provided for all the results shown in the figures and tables. I definitely want to acknowledge that a large amount of effort was no doubt devoted to producing this substantial review of the literature (with 32 studies, analyzing data from 1458 paediatric patients). However, in my opinion, the manuscript is very comprehensive but not critical, it requires some fine tuning especially in the discussion and conclusion sections to reach a good standard. Currently, the discussion section reads like a draft manuscript where facts are combined, but little interpretation is provided. In order to really make a new and helpful contribution, the authors need to develop their review by, for example, extracting more overall conceptual themes, or providing more critical assessment of either previous conclusion (from previous reviews), or the state of the literature as a whole. The lack of consistently supported concepts in the manuscript (perhaps in the literature itself) means that a reader would have a very difficult time remembering, even digesting, this dense combination of findings. It might be possible to increase the potential impact substantially, but strategic revisions are needed.

More detailed comments follow:

  1. The authors must maintain consistency of facts throughout the manuscript, such as, there is a discrepancy in the dates mentioned in the below two statements:

Page 1, Line 16 - 17: “We searched PubMed and Embase between December 2019 to March 2021 and included only peer-reviewed clinical studies or case series.”

and

Page 2, Line 69 - 70: “PubMed and Embase were searched for published papers from December 2019 to February 2021…….”

  1. Mentioning the names of authors is not necessary in the following line:

Page 2, Line 85: “Two researchers (Chen and Lo) independently assessed the eligibility of these articles.”

  1. The sentence is incomplete / unclear here:

Page 4, Line 127 – 129: “However, the studies conducted by Riphagen, Diorio, Verdoni, and Campbell included not only MIS-C patients [14-17].”

  1. There is a typo on Page 4, Table 1, Rekhtman [26], Oct 24, 2021,

This future date is incorrect and should be corrected with the actual date of this publication.

Correct this on Page 4, Table 1 and in all consecutive tables/figures accordingly throughout the manuscript.

Also, check and verify publication dates for all 32 studies mentioned in table 1 and elsewhere throughout the manuscript.

Does all 32 author names in the list (table 1 and all other tables and figures) coincides with 32 studies mentioned in the reference list? For example, I was unable to find “Rekhtman” in the reference list. This must be checked and confirmed.

Also, I could not find all the studies mentioned in the reference list to be listed on Pubmed as claimed by the authors on Page 1, Line 16 – 17 and Page 2, Line 69 - 70. Please clarify.

  1. Figure3, page7: The data in this figure represents the overall prevalence of fever “F” and gastrointestinal “G” symptoms. The overall CI for fever and gastrointestinal s/s could be better represented in the column titled “Group by systemic s/s” by replacing the final/last “F” in the column by “overall CI for F” and final “G” by “overall CI for G” or with something similar that is more convenient to the authors for clarity and ease of readability.

Similar modifications can be made for N (neurological symptoms), R (rash) and S (shock) in Figure4, page8.

Similar modifications can also be made for N (nasopharyngeal RT-PCR), and S (serum antibody) in Figure7, page11.

Also, it is a good idea to replace “N” and “S” denoting “Nasopharyngeal RT-PCR” and “Serum antibody” respectively in Figure7 with other symbols or denotations as the same denotations were used for “neurological symptoms” and “shock” in Figure4 which can be confusing for the reader.

  1. Page12: Can the authors provide any logical reasoning or inference as to why such high heterogeneities were observed in funnel plots for proportion of exposure history (88.9%), complete Kawasaki disease (80.8%) and relatively high heterogeneity observed for proportion of patients with conjunctivitis (I 2 = 63.7%)?

Perhaps, in my opinion it is highly likely that the inclusion criteria that is currently being used globally to diagnose MIS-C are too broad and it includes/overlaps children with different diseases such as Kawasaki disease (KD), toxic shock syndrome (TSS), and macrophage-activation syndrome (MAS).

  1. Page 14: Although it is interesting to note higher frequency of Kawasaki like clinical symptoms in MIS-C than in COVID-19, It would have been helpful to the readers if the authors could include additional reasoning/discussion on the male predominance in MIS-C and Kawasaki disease, or perhaps some inference drawn in regards to any underlying mechanisms for selective prevalence of similar clinical patterns of Kawasaki disease in only 30% of MIS-C patients.

  1. I would recommend the authors to rephrase the following sentence in the discussion section on Page 14, Lines 233 - 234: “Although MIS-C and Kawasaki disease have overlapping characteristics, they are distinct in several aspects”.

  1. Page 15: Conclusion section is basically the repetition of discussion and abstract. The authors might want to expand a bit more here on early prognosis, management, advanced clinical/therapeutic treatment methods of MIS-C patients associated with COVID-19.

Round 2

Reviewer 1 Report

My suggestions have been answered correctly.

Reviewer 2 Report

I have now completed the review the of the revised version of the manuscript submitted by the authors.

The authors have addressed to all my comments and concerns from round1 and have revised various points accordingly throughout the manuscript. 

I believe the manuscript has been significantly improved .